# Total and specific immunoglobulin E in induced sputum in allergic and non-allergic asthma

**Astrid Crespo-Lessmann**[ORCID]*, **Elena Curto, Eder Mateus, Lorena Soto, Alba García-Moral, Montserrat Torrejón, Alicia Belda, Jordi Giner, David Ramos-Barbón, Vicente Plaza**

Department of Respiratory Medicine and Allergology, Hospital de la Santa Creu i Sant Pau, Institute of Sant Pau Biomedical Research (IBB Sant Pau), Department of Medicine of Universitat Autònoma de Barcelona (UAB), CIBER of Respiratory Diseases (CIBERES), Barcelona, Spain

* acrespo@santpau.cat

## Abstract

### Background

Most patients with nonallergic asthma have normal serum immunoglobulin E (IgE) levels. Recent reports suggest that total and aeroallergen-specific IgE levels in induced sputum may be higher in nonallergic asthmatics than in healthy controls. Our objective is to compare total and dust-mite specific (Der p 1) IgE levels in induced sputum in allergic and nonallergic asthmatics and healthy controls.

### Methods

Total and Der p 1-specific IgE were measured in induced sputum (ImmunoCAP immunoassay) from 56 age- and sex-matched asthmatics (21 allergic, 35 nonallergic) and 9 healthy controls. Allergic asthma was defined as asthma with a positive prick test and/or clinically-significant Der p 1-specific serum IgE levels.

### Results

Patients with allergic asthma presented significantly higher total and Der p 1-specific serum IgE levels. There were no significant between-group differences in total sputum IgE. However, Der p 1-specific sputum IgE levels were significantly higher (p = 0.000) in the allergic asthmatics, but without differences between the controls and nonallergic asthmatics. Serum and sputum IgE levels were significantly correlated, both for total IgE (rho = 0.498; p = 0.000) and Der p 1-specific IgE (rho, 0.621; p = 0001).

### Conclusions

Total IgE levels measured in serum and induced sputum are significantly correlated. No significant differences were found between the differents groups in total sputum IgE. Nevertheless, the levels of Der p 1-specific sputum IgE levels were significantly higher in the allergic asthmatics, but without differences between the controls and nonallergic asthmatics. Probably due to the lack of sensitivity of the test used, but with the growing evidence for local

**Data Availability Statement:** The dataset supporting the conclusions of this article is available for consultation at www.figshare.com (DOI 10.6084/m9.figshare.11499162.v1).

**Funding:** Funding sources: Leti Grant. Fundació Catalana de Pneumologia, 2016. The funders had no role in study design, data collection and analysis, decision to publish, or preparation of the manuscript

**Competing interests:** AC. in the last three years received honoraria for speaking at sponsored meetings from AstraZeneca, Chiesi, Esteve Laboratories, Faes Farma, Ferrer, GlaxoSmithKline, Novartis, Teva, Zambon. Received help assistance to meeting travel from Bial, Novartis. Act as a consultant for AstraZeneca, Boehringer, GlaxoSmithKline, Novartis. And received funding/grant support for research projects from a variety of Government agencies and not-for-profit foundations, as well as AstraZeneca. EC has received funding to travel to and attend training activities from ALK, Menarini, Teva, AstraZeneca, Chiesi, Boehringer, and Novartis. LS, EM and declare no conflict of interests. JG has received funding to travel and attend to training activities from Menarini, Teva, AstraZeneca, Chiesi, GSK, Mundipharma, Boehringer. In the last three years, VP has received honoraria for speaking at sponsored meetings from AstraZeneca, Boehringer-Ingelheim, Chiesi, GSK, and Novartis. VP has also received financial support to travel to meetings organized by Chiesi and Novartis. VP is a consultant for ALK, AstraZeneca, Boehringer, MundiPharma, and Sanofi. VP has also received funding/grant support for research projects from a variety of governmental agencies and not-for-profit foundations, as well as from AstraZeneca, Chiesi and Menarini.

allergic reactions better methods are need to explore its presence. The Clinical Trials Identifier for this project is NCT03640936.

## Introduction

Although the causes of allergic asthma are well-understood, much less is known about the pathophysiology of nonallergic asthma. Non-allergic asthma is generally defined as nonatopic asthma with or without normal serum levels of immunoglobulin E (IgE) antibodies. Unlike allergic asthma, which is triggered by a specific allergen, nonallergic asthma may be triggered by a wide range of factors, including smoke, viruses, and other nonspecific stimuli[1].

Despite their differences, these two types of asthma may share many similarities, including airway inflammation with eosinophilia, increased Th2 cytokine production (interleukin [IL]-4, IL-5, and IL-13), hyperreactivity, and airway-induced exacerbations. Although most patients with nonallergic asthma present normal total serum IgE levels, in some cases IgE may be elevated when compared to healthy controls [2,3], with some reports suggesting that approximately 30% of asthmatic patients with a negative skin prick test have high total circulating IgE (>150 U/mL) [4,5]. These shared features suggest that unidentified environmental allergens could be involved in "nonallergic" asthma, causing a local allergic reaction in these patients[6]. Mouthuy et al. previously demonstrated that patients with nonallergic asthma present elevated levels of total IgE and *Dermatophagoides pteronyssinus* (Der p 1) specific IgE in induced sputum versus healthy controls[7]. Those findings suggest that nonallergic asthma patients may present localized allergic inflammation. This hypothesis is further supported by several studies that have demonstrated symptom relief after the administration of omalizumab—an anti-IgE treatment—in nonatopic patients[8–10]. The positive effect of omalizumab in these patients could be explained by the presence of unidentified allergens[11]. Additional support for this hypothesis comes from the relatively recent identification of a new phenotype of rhinitis—denominated local allergic rhinitis (LAR)—in patients with chronic rhinitis [12–16], characterized by local production of specific IgE with a nasal cellular Th2 immune response to nasal allergen provocation test but with negative skin prick test and undetectable serum IgE. Nevertheless, more research is needed to confirm the existence of this potential new asthma phenotype characterized by a local allergic reaction.

In this context, we hypothesized that patients with nonallergic asthma would present higher levels of IgE in induced sputum than healthy controls. To test this hypothesis, we measured total IgE and Der p 1-specific IgE in serum and induced sputum in three different groups: 1) patients with a confirmed diagnosis of allergic asthma, 2) patients diagnosed with nonallergic asthma, and 3) healthy controls. Secondary aims were to assess the correlation between total and Der p 1-specific IgE levels in serum and induced sputum and to establish a preliminary estimate of total IgE and Der p 1-specific IgE in the induced sputum of healthy individuals.

## Materials and methods

### Study design and participants

This was a comparative cross-sectional study to measure and compare total IgE and Der p 1-specific IgE levels in the induced sputum of asthmatics and in a group of healthy volunteers. Patients and controls were matched for age, sex, and disease severity; and asthma control for allergic and non-allergic groups. Patients were consecutively enrolled from the outpatient

asthma unit of our institution, a tertiary referral university hospital in Spain, between January and December 2013.

Total and Der p 1-specific IgE were measured in both serum and induced sputum. The IgE levels in serum and sputum were compared to determine the correlation between IgE levels in these two fluids.

## Definition of allergic and nonallergic asthma

We defined asthma as a history of variable respiratory symptoms and evidence of variable expiratory airflow limitation. All patients had a positive bronchodilator test or a document positive methacholine challenge test. Asthma severity was defined according to the Global Initiative for Asthma Management (GINA)[17].

Allergic asthma was defined as asthma with 1) positive skin prick test to aeroallergens and/ or 2) clinically-significant Der p 1-specific IgE according to the recommendations of the Committee of Skin Tests of the European Academy of Allergy and Clinical Immunology (EAACI) international task force[18]. Patients sensitized to various allergens were included only if dust mite allergy was the only clinically relevant one; if they showed symptoms in relation to other exposures they were excluded. Nonallergic asthma was defined as asthma with: 1) negative prick test, 2) negative Der p 1-specific IgE in serum; and 3) negative Phadiatop test (Immuno-CAP immunofluoroassay; Phadia ThermoFisher Scientific)[19]. *D. pteronyssinus* was the selected allergen because is the perennial allergen more prevalent in our geographic area.

## Inclusion and exclusion criteria

Inclusion criteria were: 1) age 18–70 years; 2) continuous residence ($>$ 4 years) in the geographic region of the study; 3) diagnosis of stable bronchial asthma according to GINA criteria [17]; 4) non-smoker; 5) no respiratory infections in the month prior to enrolment; 6) no oral corticosteroids in the last month; 7) no biological treatment with anti-IgE monoclonal antibodies; 8) no allergenic immunotherapy.

Exclusion criteria were: 1) pregnancy; 2) moderate to severe active alcohol use; 3) severe atopic dermatitis; 4) presence of any lung disease, autoimmune disease or systemic inflammatory disease, or cancer.

## Control group

The control group consisted of healthy, non-smoking volunteers age 18 to 79 years, without rhinitis, allergic asthma, or other allergic symptoms (GINA criteria) or other lung disease. Controls were recruited from among staff members at our hospital. Participation was completely voluntarily. All controls were required to present a negative prick test for aeroallergens and Der p 1-specific IgE, and negative Phadiatop test.

## Assessments and study procedures

Upon enrolment, demographic and clinical variables were assessed and recorded for all participants. On the same day, the following assessment were performed: $Fe_{NO}$ (exhaled nitric oxide test); forced spirometry; inflammatory cell count in induced sputum; eosinophil count in peripheral blood; total serum IgE levels; and skin prick test for common aeroallergens. Patients also completed the validated Spanish-language version of the Asthma Control Test (ACT)[20].

$Fe_{NO}$ was measured before spirometry using an electrochemical equipment (NO Vario Analyzer; FILT Lungen and Thorax Diagnostic GmbH, Berlin, Germany) and an expiratory maneuver providing a sustained 50 mL/s flow from total lung capacity, following the 2005

recommendations of the American Thoracic Society/European Respiratory Society[21]. Spirometry was performed using a Daptospir-600 spirometer (Sibelmed, S.A., Barcelona, Spain) in accordance with the 2003 recommendations of the Spanish Society of Pneumology and Thoracic Surgery (SEPAR), with $FEV_1$ (forced expiratory volume in 1 second) in the reference range when 80% of the predicted value [22,23].

Skin prick testing was performed according to standard procedures, with wheal diameters $\geq 4$ mm considered positive. Allergic asthma was defined as the presence of asthma symptoms for one or more allergens, with positive skin prick tests for these allergens. Well-controlled asthma was defined as ACT $\geq 20$.

### Induction and analysis of sputum

Induced sputum was evaluated following the procedures previously described by our group [24] and by Pizzichini et al.[25]. Briefly, mucus plugs were manually selected and weighed, and incubated at room temperature for 15 min in 4 times the weight (in ml) of the selected plug (in mg) in 0.1% dithiothreitol (Calbiochem, San Diego, Calif., USA), washed with 4 times the plug weight (in ml) in Dulbecco's PBS, and gravity filtered through a 41-μm-pore nylon net filter (Millipore Inc; Billerica, Mass., USA). Each specimen was homogenised and aliquoted into two equal volumes. A Neubauer hemocytometer was used to determine total cell count; visually identifiable squamous epithelial cells were not included in the total cell count. Samples with insufficient sputum cell numbers ($< 1000 \times 10^6$ cells/g) were excluded. Cell viability was determined by light microscopic assessment using trypan blue exclusion staining. The cells underwent centrifugation to obtain a cell pellet and a supernatant. The cell pellet was used for differential cell counts (macrophages, eosinophils, neutrophils, lymphocytes, and bronchial epithelial cells) performed on May-Grünwald-Giemsa-stained preparations. A differential leukocyte analysis of nonsquamous cells (Diff-Quik stained) was performed on a minimum of 400 cells. Differential cell counts are expressed as the percentage of total nonsquamous nucleated cells. Reference values for the cell counts were performed as described in other publications[26].

### Measurement of total and specific IgE antibodies

Total and specific IgE in induced sputum supernatants were measured by the ImmunoCAP fluoroenzyme immunoassay (Phadia ThermoFisher Scientific) following the manufacturer's instructions. The test was considered positive at > 2kU/L for total IgE and > 0.35kU/L for specific IgE, according to manufacturer's recommendation.

### Statistical analysis

Categorical variables were expressed as absolute and relative frequencies and quantitative variables as mean and standard deviation (SD). Groups were compared using ANOVA or chi-square test as appropriate. Given that the distribution of total and specific IgE values in serum and sputum were not normal, non-parametric tests (Mann-Whitney or Kruskal-Wallis for independent samples) were applied; the values were expressed as medians with interquartile ranges and ranges. The significance values in the case of non-parametric tests were adjusted by the Bonferroni correction. For the correlation analyses, Spearman's Rho test was used. Statistical significance was set at $p < 0.05$. Statistical analysis was performed with Statistical Package for the Social Sciences version 18.0 (SPSS, Chicago, IL).

### Ethics approval and consent to participate

The study design complied with the principles of the Declaration of Helsinki and was approved by the Clinical Research Ethics Committee at the Hospital de la Santa Creu i Sant Pau in Barcelona (COD 26/2012). All participants provided written informed consent. All patient-related data were anonymised, with the identity of the participating known only by the treating physicians. The clinicaltrials.gov identifier is NCT03640936.

## Results

A total of 56 asthmatic patients—21 (37.5%) with allergic asthma and 35 (62.5%) with nonallergic asthma—met all inclusion criteria and completed the study. The control group consisted of nine healthy volunteers. Table 1 shows the clinical, functional, and inflammatory characteristics of the patients and controls.

There were no significant differences between the three groups with regard to age, sex, or body mass index (BMI). There were significant differences between the groups in: FEV1 and need for more than one round of oral corticosteroids last 12 months. $FEV_1$ values in the allergic (83.4%) and nonallergic (80%) patient groups were similar, but substantially higher (100.2%) in the healthy controls (p = 0.013) and the need for more than one round of oral corticosteroids was significantly higher (p = 0.022) in the nonallergic group. As for other not clinically relevant sensitizations detected by prick test in the group of patients with allergic asthma, the most frequent were polysensitized patients (17), sensitized to D. farinae (8) and pet epithelia (4). A single patient had a positive prick test for molds.

Table 2 shows the total and Der p 1-specific levels of IgE in serum and sputum for all three groups. Significant between-group differences in serum IgE levels were observed for both total and Der p 1-specific IgE, with the allergic asthma group presenting significantly higher levels of total IgE (mean, 1702.3 KU/L) and Der p1-specific IgE levels (15.5 KU/L) than the nonallergic group and healthy controls. In the sputum samples, no significant between-group differences in total IgE were observed. However, Der p 1-specific IgE levels were significantly higher

**Table 1. Clinical, functional, and inflammatory characteristics of patients and controls.**

| Variables | Nonallergic asthma (n = 21) | Allergic asthma (n = 35) | Healthy controls (n = 9) | P |
|---|---|---|---|---|
| Age (years) | 54.8 (14.8) | 51.7(13.6) | 41.88 | 0.076 |
| Sex (% females) | 57.1% | 51.4% | 88.8% | 0.125 |
| BMI (kg/m$^2$) | 28.7 (4.1) | 27.3 (5.9) | 25.5 (6.48) | 0.353 |
| $FEV_1$ (%) | 83.4 (13.9) | 80 (20.8) | 100.2 (10.74) | 0.013 |
| Serum eosinophils (x10$^9$/L) | 0.302 (0.310) | 0.271 (0.239) | 0.095 (0.068) | 0.113 |
| Sputum eosinophils (%) | 10.33 (19.54) | 11.2 (13.01) | 0.77 (1.09) | 0.166 |
| Rhinitis | 61.9% | 82.8% | n/a | 0.080 |
| Nasal polyposis (%) | 28.5% | 20% | n/a | 0.462 |
| Severe persistent asthma (%) | 42.8% | 42.8% | n/a | 0.870 |
| GINA 4.0 scale, grade 5–6 (%) | 52.4% | 48.5% | n/a | 0.590 |
| Good asthma control (ACT >20%) | 19% | 20% | n/a | 0.357 |
| Emergency room visits last 12 months (%) | 28.5% | 11.4% | n/a | 0.182 |
| >1 round of oral corticosteroids last 12 months | 38% | 8.5% | n/a | 0.022 |
| Inflammatory phenotype in induced sputum (%) | Paucigranulocytic: 28.6% | Paucigranulocytic: 22.4% | n/a | 0.425 |
| | Mixed: 9.5% | Mixed: 8.6% | | |
| | Eosinophilic: 33.3% | Eosinophilic: 54.3% | | |
| | Neutrophilic: 28.6% | Neutrophilic: 14.3% | | |

**Table 2. Total and dust-mite specific IgE n serum and sputum in patients with allergic asthma, non-allergic asthma, and healthy controls.**

| VARIABLE | NONALLERGIC ASTHMA (N = 21) | ALLERGIC ASTHMA (N = 35) | HEALTHY CONTROLS (N = 9) | p |
|---|---|---|---|---|
| High quality induced sputum (%)* | 61.9% | 69.7% | 44% | 0.294 |
| Total IgE (KU/L) in serum, median (IQR) | 55.5 (177.93) | 233.5 (368.25) | 14.25 (38.30) | <0.0001 |
| Total IgE (KU/L) in serum, average range | 423.85 | 1702.3 | 87.78 | |
| *Der p*-specific IgE (KU/L) in serum, median (IQR) | 0.01 (0.03) | 15.45 (32.68) | 0.02 (0.04) | <0.0001 |
| *Der p*-specific IgE (KU/L) in serum, average range | 0.08 | 99.97 | 0.10 | |
| Total IgE (kU/L) in sputum, median (IQR) | 2.69 (4.55) | 4.5 (2.18) | 3.16 (1.41) | 0.188 |
| Total IgE (kU/L) in sputum, average range | 8.11 | 7.45 | 4.12 | |
| *Der p*-specific IgE in sputum (kU/L), median (IQR) | 0.055 (0.03) | 0.095 (0.09) | 0.06 (0.02) | <0.0001 |
| *Der p*-specific IgE in sputum (kU/L), average range | 0.04 | 0.54 | 0.05 | |

*High quality defined as > 40% viability, <20% epithelial cells, > $1x10^6$ cells

(p<0.0001) in the allergic asthma group compared to the other two groups. **Fig 1** shows the total sputum IgE levels in the three groups, with no significant differences between the groups.

**Fig 2** shows the specific Der p 1-specific IgE levels in sputum for each group. The allergic asthma group presented significantly higher Der p 1-specific IgE levels than both the nonallergic patients (p<0.0001) and the healthy controls (p = 0.006). There were no significant differences between the nonallergic patients and the healthy controls. In the group of patients with non-allergic asthma, a subgroup analysis was performed among those with total sputum IgE >2 or <2, with no statistically significant differences either at clinical characteristics or regarding the other IgE measurements.

## Correlations

Using data from the whole sample (patients and controls), we calculated the correlations between total and Der p 1-specific IgE levels in sputum and serum. **Table 3** shows a matrix with all significant correlations (p<0.0001) in descending order from strongest to weakest. As that table makes clear, the strongest correlation was between total IgE and Der p 1-specific IgE in serum.

## Discussion

In the present study, we sought to test the hypothesis that the airways of patients with nonallergic asthma exhibit a local inflammatory response by determining total and dust mite-specific IgE antibody levels in induced sputum. Our results showed that patients with allergic asthma presented significantly higher total and Der p 1-specific IgE levels in serum compared to both nonallergic asthmatics and healthy controls. However, contrary to our expectations, there were no significant differences among the three groups in total sputum IgE levels. Moreover, there were no differences in Der p 1-specific sputum IgE levels between healthy controls and nonallergic asthma patients. Overall, the lack of significant differences in total sputum IgE levels between the three groups was surprising. However, diverse factors could explain this unexpected finding, as we discuss in detail below.

The hypothesis that patients with nonallergic asthma may present a local allergic response in the airways derives from the mounting evidence for a new phenotype of local allergic

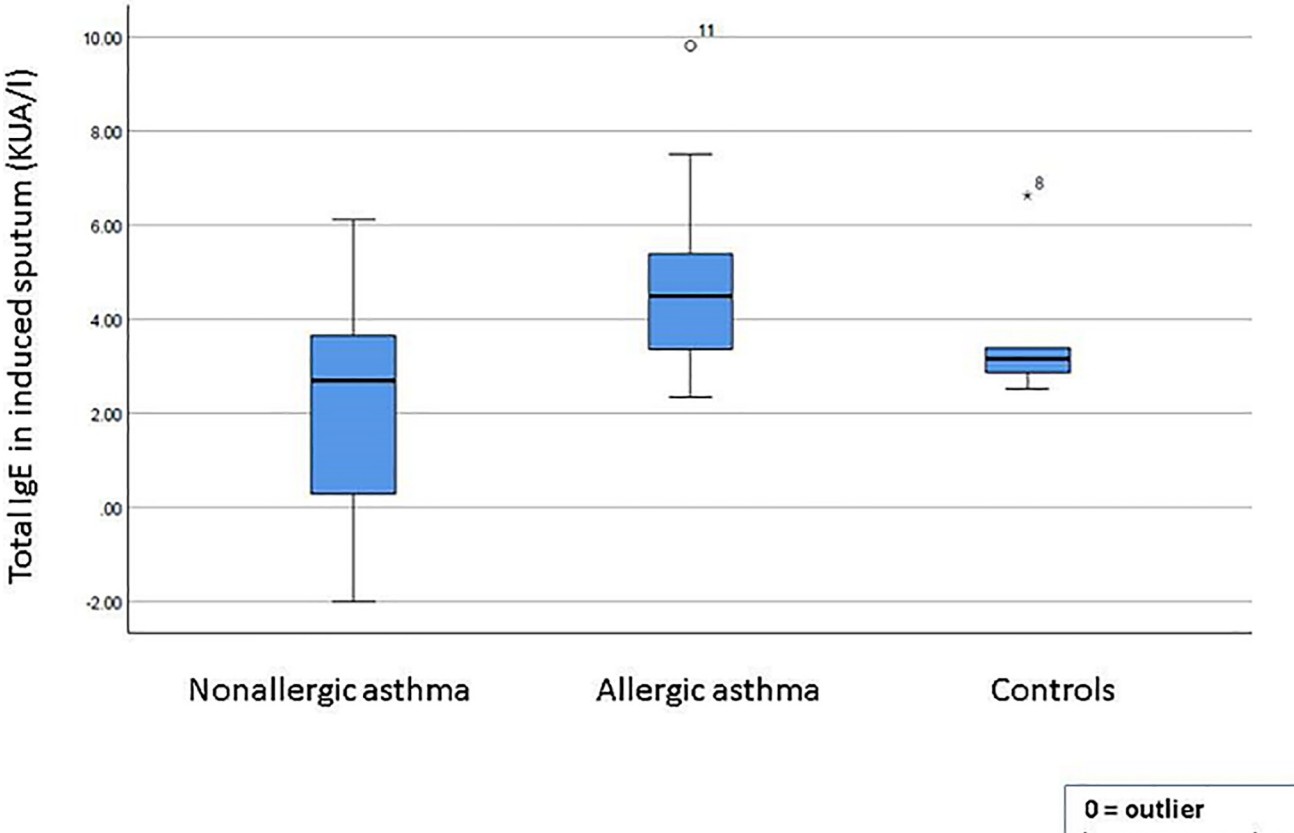

**Fig 1. Total IgE in sputum in patients with allergic asthma, non-allergic asthma, and healthy controls.**

rhinitis in nonatopic rhinitis[12–15]. Studies have demonstrated local production of specific IgE in the nostrils of patients with negative skin prick test and undetectable serum IgE[12]. Given the similarities between asthma and rhinitis, it seems highly plausible that patients with nonatopic asthma could also present a local allergic response, particularly considering the lack of a clear physiopathologic explanation for nonallergic asthma. Although there are differences between allergic and nonallergic asthma, these two clinical entities share many similarities, including airway inflammation with eosinophilia, increased Th2 cytokine production, airway-induced exacerbations[5,27]. In addition, up to 30% of patients with nonallergic asthma may present elevated total serum IgE levels[4,5,16]. These shared features suggest that unidentified environmental allergens—which stimulate a local allergic reaction—may be responsible for the symptoms experienced by patients with nonallergic asthma, a hypothesis supported by a small but growing body of evidence showing local airway synthesis of IgE[27], even in patients without any allergen-specific serum IgE[7]. Moreover, the findings from multiple studies that the anti-IgE treatment omalizumab provides symptom relief in nonatopic patients implies that an inflammatory reaction does, in fact, play a role in these patients[8,9,10]. Mouthuy et al.[7] found that nonallergic asthmatics present elevated levels of total and Der p 1-specific IgE in induced sputum compared to healthy controls, a finding that supports the concept of local airway inflammation in those patients. However, we were unable to confirm those findings, as we found no significant differences between the nonallergic asthmatics and healthy controls in IgE levels (both total and Der p 1-specific) in induced sputum or in serum. Our findings were

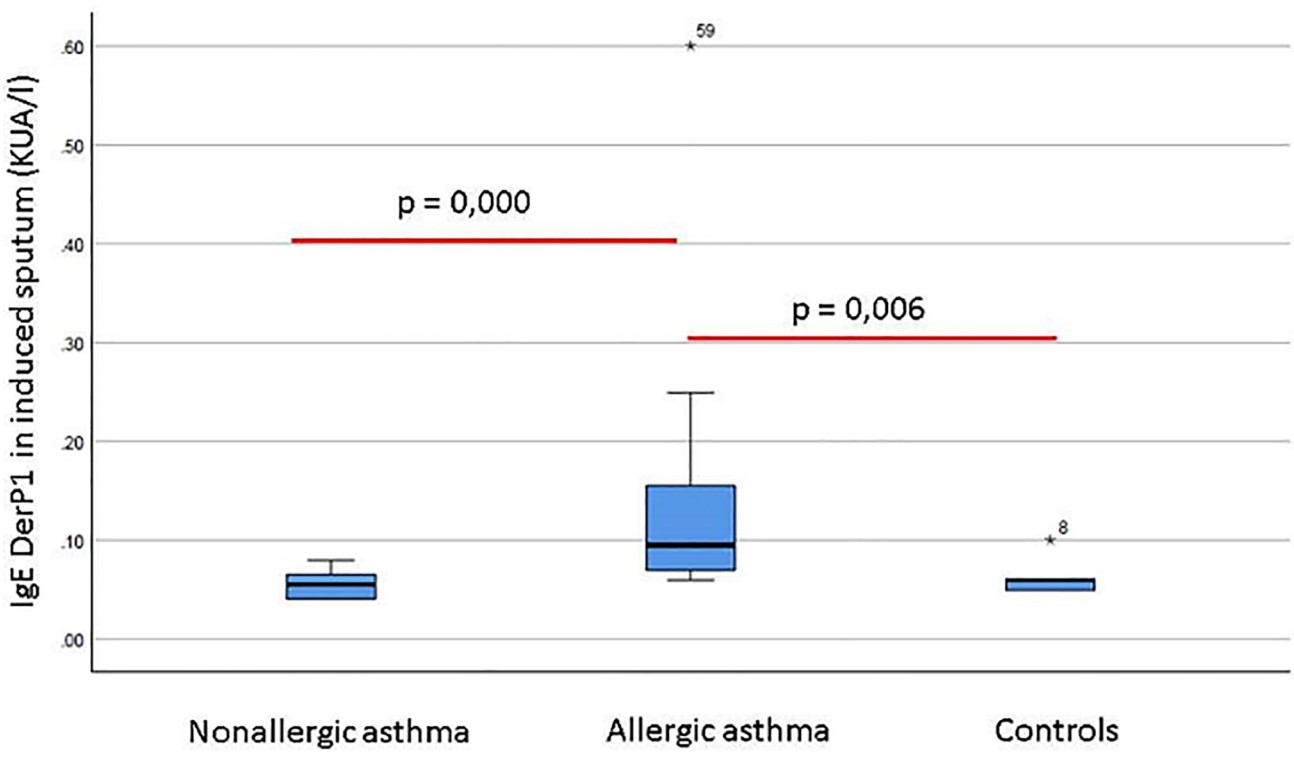

**Fig 2. Der p 1-specific IgE levels in sputum in patients with allergic asthma, non-allergic asthma and healthy controls.**

closer to those reported by Manise et al.[28], who—in contrast to Mouthuy et al.—found that total sputum IgE levels were higher in atopic than in nonatopic asthmatics and that there were no differences between nonatopic asthmatics and nonatopic healthy subjects. In view of the heterogeneous findings of these three studies, it is clear that more work will be needed to clarify whether or not there are truly differences among nonatopic and atopic asthmatics and healthy controls with regard to IgE levels in sputum.

There are several reasons that could explain the discrepancy between our results and those reported by Mouthuy et al. First, the lack of any significant differences in IgE levels between

**Table 3. Significant correlations between IgE values in serum and induced sputum.**

| Serum | | Sputum | | Correlation† |
|---|---|---|---|---|
| **Total IgE** | **_Der-p_ IgE** | **Total IgE** | **_Der-p_ IgE** | |
| X | X | | | 0.658 |
| | X | | X | 0.621 |
| X | | | X | 0.538 |
| X | | X | | 0.498 |
| | | X | X | 0.454 |

†Spearman's rho; P = 0.000 for all correlations

the nonallergic asthmatics and healthy controls in our study could potentially be attributed to the size of the control group in our study (n = 9) versus the large control group (n = 25) in the study by Mouthy and colleagues. Another explanation could be related to the ImmunoCAP method, which may not be sufficiently sensitive to detect very small differences in IgE levels. In our sample, although the test did detect higher Der p 1-specific IgE levels in the sputum of allergic asthmatics verus both nonallergic asthmatics and healthy controls, this is probably because there were large differences in Der p 1-specific IgE levels in the allergic asthmatics versus the nonallergic asthmatics and healthy controls given that the study inclusion criteria for those latter two groups specifically required that they have a negative skin prick and negative serum Der p 1. By contrast, the differences in total IgE may have been less marked, making it more difficult to detect using the ImmunoCap technique.

Local IgE production has been documented previously in nonallergic asthma patients using bronchial biopsy[16,29,30]. In this regard, the lack of a significant difference between the three groups in our study with regard to total sputum IgE was surprising, especially considering that —at the very least—allergic asthmatics would be expected to present substantially higher IgE levels than healthy controls. Although the reason for this unexpected finding is not clear, it could be due to the relatively small sample size or to the limited capacity of the ImmunoCap technique to detect small differences in total IgE. In this regard, larger studies will be needed, perhaps using alternative methods to measure sputum IgE.

## Correlation between IgE in sputum and serum

In the present study, we used the sputum induction and analysis methods described by Araujo et al.[31], who validated laboratory measurements of total and Der p 1-specific IgE in induced sputum supernatant versus serum levels, but in a heterogeneous sample involving patients with diverse pathologies, not only asthma. In recent years, several studies have correlated total IgE levels in induced sputum and serum, with sometimes contradictory findings[32]. In the present study, we found a highly significant correlation (rho, 0.498; p = 0.000) between total IgE levels in serum and sputum in our overall sample, a finding that is consistent with the results described by Manise et al. in asthmatic patients[28]. By contrast, other authors, including Mouthuy et al.[7] and Ahn et al[33] have not found any correlation between total IgE levels in sputum and serum. These contrasting findings raise further doubts about the sensitivity of the ImmunoCap technique used to measure IgE in sputum, potentially providing an additional explanation for the differences between our results and those of Mouthuy et al.

Data from a recent study conducted by Pillai et al.[34] could help to explain why we were unable to detect significant between-group differences in total IgE in induced sputum. Those authors suggest that IgE produced in the bronchial mucosa of nonatopic asthmatic patients may remain confined to the mucosa, bound to cells that carry those receptors. If this hypothesis is correct, it would explain why IgE is not readily detectable in induced sputum. Pillai and colleagues posited that both atopic and nonatopic asthmatics would have greater total IgE concentrations in the airways than in serum. To test this, they determined IgE levels in the blood and bronchial mucosa of 10 atopic and 10 nonatopic asthmatic patients and 10 nonatopic controls, finding that median total IgE levels were significantly elevated in both the atopic and nonatopic asthmatic patients versus controls. These data are consistent with the hypothesis that IgE synthesis, sequestration, or both are ongoing in the bronchial mucosa of both nonatopic and atopic asthmatic patients. Interestingly, Pillai et al. also suggest, as other several authors have previously proposed[10,35,36], that increased bronchial mucosal IgE production in nonatopic asthmatic patients may be directed against targets other than allergens, including

possible "autoallergens", or that there are allergen-independent roles for IgE in the pathophysiology of asthma.

Overall, the findings reported in studies which use more invasive but more sensitive methods (e.g., bronchial biopsy) strongly suggest the presence of elevated total IgE production in the airways of both allergic and nonallergic asthma patients[10,34–36]. The fact that we were unable to confirm the findings reported by Mouthuy et al. with regard to detecting significant differences in sputum IgE levels between nonatopic asthmatics and healthy controls, together with the contradictory data reported to date regarding the correlation between IgE levels in serum and induced sputum, suggests that more sensitive methods of measuring IgE in sputum may be required.

### Total and Der p-specific IgE in induced sputum in healthy individuals

A secondary aim of this study was to determine, on a pilot basis, the standard levels of total and Der p 1-specific IgE in the induced sputum of healthy controls. We found the following values: median (IQR) total IgE in sputum in the controls was 3.16 (1.41) KUA/L. The Der p 1-specific values were 0.06 (0.02). Evidently, the small sample (n = 9) of healthy controls are insufficient to define standard levels, but these data provide an initial estimate. Nevertheless, more data from larger studies will be needed to confirm these initial levels, especially considering the small sample and the potential limitations in the assay technique used to measure IgE in the sputum supernatant.

### Study strengths and limitations

The main limitation of the present study is the limited number of healthy controls. Another limitation may be related to the lack of significant differences between the groups in total IgE in induced sputum, which points to limitations in the measurement technique that may have influenced our findings. An important strength of the study is that this is, to our knowledge, only the second study conducted to date to specifically determine total and Der p-specific IgE in induced sputum of allergic, nonallergic, and healthy controls. Given the conflicting results of our study and those reported by Mouthuy et al., additional studies are needed. Finally, another important strength is the well-selected and well-defined sample of patients and controls; we used very strict diagnostic criteria (based on the most recent clinical guidelines) to define both allergic and nonallergic asthma, as well as for the healthy controls.

### Conclusions

The findings of this study show that total IgE levels measured in serum and induced sputum are significantly correlated. The significantly higher levels of Der p 1-specific IgE detected in the induced sputum of the allergic asthmatics underscores the potential value of measuring aeroallergen-specific IgE in induced sputum. Nevertheless, the lack of significant between-group differences in total sputum IgE levels suggests that the ImmunoCAP immunoassay technique used in this study may not be sufficiently sensitive to detect small differences in total sputum IgE. To support the results of this work, a larger sample size would be necessary for future studies.

A growing body of evidence indicates that both allergic and nonallergic asthmatics present elevated airway inflammation. Measuring IgE levels in induced sputum is a non-invasive, cost-effective approach that could provide valuable clinical data to help individualize the treatment of nonallergic asthma. However, more sensitive methods are needed to measure IgE levels in induced sputum. Nonetheless, there exists a clear potential to use total and/or aeroallergen-specific IgE levels measured in induced sputum as a marker of treatment efficacy.

## Acknowledgments

The authors wish to thank the patients and volunteers who generous contributed to this study. We also thank Bradley Londres for editing the manuscript.

## Author Contributions

**Conceptualization:** Lorena Soto, Alba García-Moral, David Ramos-Barbón, Vicente Plaza.

**Data curation:** Elena Curto, Lorena Soto, Alba García-Moral, Vicente Plaza.

**Formal analysis:** Astrid Crespo-Lessmann, Vicente Plaza.

**Investigation:** Astrid Crespo-Lessmann, Elena Curto, Eder Mateus, Lorena Soto, Alba García-Moral, Montserrat Torrejón, Alicia Belda, Jordi Giner, David Ramos-Barbón.

**Methodology:** Astrid Crespo-Lessmann, Elena Curto, Eder Mateus, Montserrat Torrejón, Alicia Belda, Jordi Giner, David Ramos-Barbón.

**Project administration:** Astrid Crespo-Lessmann, Montserrat Torrejón, Alicia Belda, Jordi Giner, Vicente Plaza.

**Software:** Elena Curto.

**Supervision:** Astrid Crespo-Lessmann, David Ramos-Barbón.

**Validation:** Astrid Crespo-Lessmann, Vicente Plaza.

**Visualization:** Vicente Plaza.

**Writing – original draft:** Astrid Crespo-Lessmann, Elena Curto.

**Writing – review & editing:** Astrid Crespo-Lessmann, Elena Curto.

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
