## [Decision Letter · Decision Letter 0]

28 Nov 2019

PONE-D-19-27179

Total and specific immunoglobulin E in induced sputum in allergic and non-allergic asthma

PLOS ONE

Dear Dr. Crespo,

Thank you for submitting your manuscript to PLOS ONE. After careful consideration, we feel that it has merit but does not fully meet PLOS ONE’s publication criteria as it currently stands. Therefore, we invite you to submit a revised version of the manuscript that addresses the points raised during the review process.

We would appreciate receiving your revised manuscript by Jan 12 2020 11:59PM. To enhance the reproducibility of your results, we recommend that if applicable you deposit your laboratory protocols in protocols.io, where a protocol can be assigned its own identifier (DOI) such that it can be cited independently in the future. For instructions see: http://journals.plos.org/plosone/s/submission-guidelines#loc-laboratory-protocols

We look forward to receiving your revised manuscript.

Kind regards,

Aleksandra Barac

Academic Editor

PLOS ONE

Journal Requirements:

2. We noticed you have some minor occurrence(s) of overlapping text with the following previous publication(s), which needs to be addressed:

https://doi.org/10.1159/000431259

https://doi.org/10.2147/JAA.S142200

http://www.jiaci.org/issues/vol25issue6/6.pdf

In your revision ensure you cite all your sources (including your own works), and quote or rephrase any duplicated text outside the Methods section. Further consideration is dependent on these concerns being addressed.

3. We note that you have reported significance probabilities of 0 in places. Since p=0 is not strictly possible, please correct this to a more appropriate limit, eg 'p<0.0001'.

5. Your ethics statement must appear in the Methods section of your manuscript. If your ethics statement is written in any section besides the Methods, please move it to the Methods section and delete it from any other section. Please also ensure that your ethics statement is included in your manuscript, as the ethics section of your online submission will not be published alongside your manuscript.

6. Thank you for stating the following in the Competing Interests section:

AC. in the last three years received honoraria for speaking at sponsored meetings from AstraZeneca, Chiesi, Esteve Laboratories, Faes Farma, Ferrer, GlaxoSmithKline, Novartis, Teva, Zambon.  Received help assistance to meeting travel from Bial, Novartis. Act as a consultant for AstraZeneca, Boehringer, GlaxoSmithKline, Novartis. And received funding/grant support for research projects from a variety of Government agencies and not-for-profit foundations, as well as AstraZeneca.

EC has received funding to travel to and attend training activities from ALK, Menarini, Teva, AstraZeneca, Chiesi, Boehringer, and Novartis.

LS, EM and declare no conflict of interests.

JG has received funding to travel and attend to training activities from Menarini, Teva, AstraZeneca, Chiesi, GSK, Mundipharma, Boehringer.

In the last three years, VP has received honoraria for speaking at sponsored meetings from AstraZeneca, Boehringer-Ingelheim, Chiesi, GSK, and Novartis. VP has also received financial support to travel to meetings organized by Chiesi and Novartis. VP is a consultant for ALK, AstraZeneca, Boehringer, MundiPharma, and Sanofi. VP has also received funding/grant support for research projects from a variety of governmental agencies and not-for-profit foundations, as well as from AstraZeneca, Chiesi and Menarini.

Reviewers' comments:

Reviewer's Responses to Questions

**Comments to the Author**

1. Is the manuscript technically sound, and do the data support the conclusions?

Reviewer #1: Partly

Reviewer #2: No

2. Has the statistical analysis been performed appropriately and rigorously? 

Reviewer #1: N/A

Reviewer #2: No

3. Have the authors made all data underlying the findings in their manuscript fully available?

Reviewer #1: Yes

Reviewer #2: Yes

4. Is the manuscript presented in an intelligible fashion and written in standard English?

Reviewer #1: Yes

Reviewer #2: Yes

5. Review Comments to the Author

Reviewer #1: Dear Editor,

The article entitled as “Total and specific immunoglobulin E in induced sputum in allergic and non-allergic Asthma” was reviewed for PLOS ONE on 20 November 2019. It is a valuable study in means of using and evaluating induced sputum and it has a control group. However there are the following points that needs to be solved;

- The size of the study group seems to be too small, this may be increased, or the power of the study may be calculated and added.

- The subject of the study that comparing total IgE between allergic and nonallergic asthma is not new even tough using induced sputum and spesicifc IgE.

- Local allergy or entopy was showed in nonallergic rhinitis or asthma in mucosa biopsy or lavage in 2000’s. However, it has lost its popularity because of the difficulty to diagnose these patients, and not changing the treatment. Furthermore, it is not evaluated as an asthma phenotype. The new point may be investigating entopy between allergic, non-allergic eosinophilic, and non-allergic non-eosinophilic phenotypes.

- Subgroup analysis as high and low total ıge may be performed.

- There are other allergens such as pollens that may be present in nonallergic astmatics sputum. This question effects the results unconfident.

- Mold sensitivity has a great impact on total IgE, and should have been explored.

Reviewer #2: The detection limit was 0.35 kU/L for specific IgE and yet the authors report levels in sputum markedly below this level, e.g. for Der p-specific IgE 0.055 kU/L median and 0.04 kU/L average for non-allergic asthma. (See line 173-4 and Table 2.) Thus unfortunately, the data appear to be invalid.

6. PLOS authors have the option to publish the peer review history of their article (what does this mean?). If published, this will include your full peer review and any attached files.

Reviewer #1: No

Reviewer #2: No

---

## [Author Response · Author response to Decision Letter 0]

3 Jan 2020

1. Is the manuscript technically sound, and do the data support the conclusions?

 Reviewer #1: Partly

 Reviewer #2: No

Author’s response: According to the recommendations of the reviewer, we have better drafted the manuscript in key points to make it more understandable, especially in the conclusions (see line 41)

2. Has the statistical analysis been performed appropriately and rigorously?

 Reviewer #1: N/A

 Reviewer #2: No

Author’s response: The statistical analysis was carried out under the support of a statistician who guided us at all times on the most appropriate analyzes for this study. If you suggest doing some analysis of greater interest in our study, using the database that we have attached to the study, we would be delighted to perform it.

3. Have the authors made all data underlying the findings in their manuscript fully available?

 Reviewer #1: Yes

 Reviewer #2: Yes

Author’s response: As required by the internal policy of Plos one, we have attached our data to a public repository. 

4. Is the manuscript presented in an intelligible fashion and written in standard English?

 Reviewer #1: Yes

 Reviewer #2: Yes

Author’s response: Thanks for your feedback

5. Review Comments to the Author

Reviewer #1: Dear Editor,

 The article entitled as “Total and specific immunoglobulin E in induced sputum in allergic and non-allergic Asthma” was reviewed for PLOS ONE on 20 November 2019. It is a valuable study in means of using and evaluating induced sputum and it has a control group. However there are the following points that needs to be solved;

 - The size of the study group seems to be too small, this may be increased, or the power of the study may be calculated and added.

Author’s response: Due to the limited literature and great variability in the results that exist of the IgE values in the induced sputum, for the calculation of the sample size, the reference values of the IgE in the induced sputum will be considered as binding variable reports in literature (references 31 and 32). This sample size is calculated with the Granmo V7.10 program, setting the type I error at the usual 5% (α = 0.05), with a bilateral approximation and with a minimum difference required for a power of 80% or higher (ß = 0.2). Also our study had as a secondary objective to establish a preliminary estimate of total IgE and Der p 1-specific IgE in the induced sputum of healthy individuals. To support the results of this work, a larger sample size would be necessary for future studies. A comment has been added in the manuscript (see line 327)

 - The subject of the study that comparing total IgE between allergic and nonallergic asthma is not new even tough using induced sputum and specific IgE.

Author’s response: The levels of total and specific IgE in blood are extensively studied comparatively in asthmatic and healthy patients, and also in different types of asthmatic patients. However, sputum-induced levels of this immunoglobulin have been poorly reported. Our particular interest was to assess its potential applicability as a diagnostic test in the detection of local allergic reactions at the bronchial level. 

- Local allergy or entopy was showed in nonallergic rhinitis or asthma in mucosa biopsy or lavage in 2000’s. However, it has lost its popularity because of the difficulty to diagnose these patients, and not changing the treatment. Furthermore, it is not evaluated as an asthma phenotype. The new point may be investigating entopy between allergic, non-allergic eosinophilic, and non-allergic non-eosinophilic phenotypes.

Author’s response: We totally agree with the reviewer's opinion. Local allergy has been a controversial issue in the last decade, and in our view further research is needed to define this entity properly. Its loss of popularity has been due, on the one hand, to diagnostic difficulty, so this study has been, in fact, a first approach to a technique that could allow diagnosing it in a simple way. On the other hand, until now it did not imply differences in treatment, but we believe that with the availability of new monoclonal drugs this may change. We also consider that it could help to better understand a group of patients, who because a priori they do not present signs of atopy or eosinophilia, remain limited to conventional treatments. 

- Subgroup analysis as high and low total IgE may be performed.

Author’s response: The main focus of our work was:

1) To measure the levels of total and Dermatophagoides pteronyssinus (d1) specific IgE in the induced sputum (IS) of asthmatic patients and healthy volunteers

2) To correlate the levels of total local IgE and specific IgE to d1 (sputum and peripheral blood) in patients with allergic and non-allergic asthma

Since there are few published papers in the literature that have measured levels of total IgE in sputum and since there are no established values of normal sputum total IgE levels, we cannot make this classification as there is no established cut-off point.

- There are other allergens such as pollens that may be present in nonallergic asthmatics sputum. This question effects the results unconfident.

Author’s response: Our inclusion criteria were particularly strict and detailed in this study. We only selected patients with allergic asthma in whom D. pteronyssinus was the only clinically relevant allergen, in addition to presenting positive prick test, and/or previous high specific IgE for this allergen. If they presented a positive prick test for other allergens and there was any suspicion that their symptoms were related to other allergens to which they were sensitized, they were excluded from the study. This has been clarified in the manuscript (see line 140)

 - Mold sensitivity has a great impact on total IgE, and should have been explored.

Author’s response: This is an interesting point of view, but in our area, dust mites are the most common perennial allergens, and mold sensitization is not particularly common. In fact, in this group of patients, only one of them had a positive prick test for Aspergillus, and it was not clinically relevant (see line 219)

Reviewer #2: The detection limit was 0.35 kU/L for specific IgE and yet the authors report levels in sputum markedly below this level, e.g. for Der p-specific IgE 0.055 kU/L median and 0.04 kU/L average for non-allergic asthma. (See line 173-4 and Table 2.) Thus unfortunately, the data appear to be invalid.

Author’s response: According to manufacturer’s recommendation, > 2kU/L for total IgE and > 0.35kU/L for specific IgE, are the values to consider a positive test, but as we previously stated, the levels in induced sputum are not described yet. We understand that the previous phrasing could be confusing, so we have modified the corresponding paragraph to better understand this aspect.

---

## [Editor Report · Decision Letter 1]

7 Jan 2020

Total and specific immunoglobulin E in induced sputum in allergic and non-allergic asthma

PONE-D-19-27179R1

Dear Dr. Crespo-Lessman,

We are pleased to inform you that your manuscript has been judged scientifically suitable for publication and will be formally accepted for publication once it complies with all outstanding technical requirements.

With kind regards,

Aleksandra Barac

Academic Editor

PLOS ONE
---

## [Editor Report · Acceptance letter]

8 Jan 2020

PONE-D-19-27179R1 

Total and specific immunoglobulin E in induced sputum in allergic and non-allergic asthma 

Dear Dr. Crespo-Lessman:

I am pleased to inform you that your manuscript has been deemed suitable for publication in PLOS ONE. Congratulations! Your manuscript is now with our production department. 

With kind regards,

on behalf of

Dr. Aleksandra Barac 

Academic Editor

PLOS ONE